# Exenatide once weekly over 2 years as a potential disease-modifying treatment for Parkinson's disease: protocol for a multicentre, randomised, double blind, parallel group, placebo controlled, phase 3 trial: The 'Exenatide-PD3' study

Nirosen Vijiaratnam [1,2] Christine Girges,[1,2] Grace Auld,[3] Marisa Chau,[3] Kate Maclagan,[3] Alexa King,[3] Simon Skene [4,5] Kashfia Chowdhury,[3] Steve Hibbert,[3] Huw Morris,[1,2] Patricia Limousin,[1,2] Dilan Athauda,[1,2] Camille B Carroll [6,7] Michele T Hu,[8,9,10] Monty Silverdale,[11] Gordon W Duncan,[12,13] Ray Chaudhuri,[14] Christine Lo,[9] Silvia Del Din,[15] Alison J Yarnall,[15,16] Lynn Rochester,[15] Rachel Gibson,[17] John Dickson,[18] Rachael Hunter,[19] Vincenzo Libri,[20,21] Thomas Foltynie [1,2]

► Prepublication history and additional online supplemental material for this paper are available online. To view these files, please visit the journal online (http://dx.doi.org/10.1136/bmjopen-2020-047993).

For numbered affiliations see end of article.

**Correspondence to**
Professor Thomas Foltynie;
t.foltynie@ucl.ac.uk

## ABSTRACT

**Introduction** Parkinson's disease (PD) is a common neurodegenerative disorder with substantial morbidity. No disease-modifying treatments currently exist. The glucagon like peptide-1 receptor agonist exenatide has been associated in single-centre studies with reduced motor deterioration over 1 year. The aim of this multicentre UK trial is to confirm whether these previous positive results are maintained in a larger number of participants over 2 years and if effects accumulate with prolonged drug exposure.

**Methods and analysis** This is a phase 3, multicentre, double-blind, randomised, placebo-controlled trial of exenatide at a dose of 2 mg weekly in 200 participants with mild to moderate PD. Treatment duration is 96 weeks. Randomisation is 1:1, drug to placebo. Assessments are performed at baseline, week 12, 24, 36, 48, 60, 72, 84 and 96 weeks.

The primary outcome is the comparison of Movement Disorders Society Unified Parkinson's Disease Rating Scale part 3 motor subscore in the practically defined OFF medication state at 96 weeks between participants according to treatment allocation. Secondary outcomes will compare the change between groups among other motor, non-motor and cognitive scores. The primary outcome will be reported using descriptive statistics and comparisons between treatment groups using a mixed model, adjusting for baseline scores. Secondary outcomes will be summarised between treatment groups using summary statistics and appropriate statistical tests to assess for significant differences.

**Ethics and dissemination** This trial has been approved by the South Central-Berkshire Research Ethics Committee and the Health Research Authority. Results will be disseminated in peer-reviewed journals, presented at scientific meetings and to patients in lay-summary format.

**Trial registration numbers** NCT04232969, ISRCTN14552789.

## Strengths and limitations of this study

► This is the protocol for the first phase 3 double-blind, randomised, placebo controlled trial of exenatide in Parkinson's disease.

► This study uses novel secondary outcome measures in substudies (cerebrospinal fluid analysis, dopamine transporter imaging and digital technology measurement devices) which should provide a more sensitive and comprehensive assessment of potential disease modification.

► Although the 2-year follow-up period should provide a more definitive signal on disease modification, this will take longer to report findings and has risks for long-term patient retention.

## INTRODUCTION

Parkinson's disease (PD) is the second most common neurodegenerative disease affecting over 10 million people worldwide and its prevalence is increasing.[1] Symptomatic treatments are available and mainly focus on dopamine replacement strategies.[2 3] Such therapies provide improvements in the core motor features of PD: tremor, limb rigidity and slowness of movement (bradykinesia).[4] These symptomatic treatments do not impact

on the progressive nature of the disease nor the majority of the non-motor symptoms (NMS). Moreover, with time, some patients will develop dopamine-refractory gait and balance problems leading to falls and risk of fractures; speech and swallowing problems leading to difficulty in communication and aspiration pneumonia, cognitive impairment, visual hallucinations and dementia with mounting care needs.[3][4] These complications result in increased dependence, caregiver strain, need for 24 hours care and death. Therefore, PD is a growing problem for individuals, healthcare and society making the development of disease modifying treatments imperative.

Exenatide (exendin-4) is a licensed and effective treatment for patients with type 2 diabetes mellitus (T2DM).[5] It is an agonist for the glucagon-like peptide 1 (GLP-1) receptor and in the presence of elevated blood glucose stimulates insulin release. It also increases pancreatic beta islet cell mass and reduces apoptosis. Exenatide has been the subject of multiple phase 3 trials in patients with type 2 diabetes and was granted a license for the treatment of type 2 diabetes in 2006.[5]

In parallel with the confirmation of the beneficial effects of exenatide on glucose control, laboratory work has showed that exenatide has beneficial effects on neurons in vitro.[6] Exenatide induces neurite outgrowth, promotes neuronal differentiation and rescues degenerating neuronal cells while also reversing neurotoxin induced damage in animal models.[6][7] These neurotrophic properties have sparked interest regarding its potential use as a neurodegenerative disease-modifying agent.[8][9]

The specific relevance of exenatide to PD has also been extensively evaluated. Exenatide has been shown to increase transcription of tyrosine hydroxylase (the rate limiting enzyme in dopamine synthesis) in brainstem catecholaminergic neurons.[10] Furthermore, stimulation of GLP-1 receptors may have beneficial effects on the neurodegenerative processes of PD through downstream cellular pathways.[6][11] These findings are further supported by a recent study suggesting a reduced future risk of developing PD in T2DM patients treated with GLP-1 agents.[12]

To investigate the potential effects of exenatide in patients with PD, an investigator-initiated pilot trial was undertaken.[13] This open-label, parallel group, randomised controlled trial evaluated the tolerability of exenatide (Byetta 10 µg two times per day) in 45 patients with moderately severe PD (Hoehn and Yahr stage of less than 2.5) over an exposure period of 48 weeks with a subsequent washout period of 12 weeks. This showed an advantage of 4.9 points in in the Movement Disorders Society Unified Parkinson's Disease Rating Scale (MDS-UPDRS) part 3 (motor subscore) in exenatide treated patients at 12 months which persisted even after a 12-week washout period. Clinically important differences in cognition were also noted. Serial DaTscan (Ioflupane I 123 injection) imaging showed no progression between baseline and 48 weeks in the exenatide treated patients.[13]

A further phase 2 double blind randomised controlled trial evaluating the effects of exenatide in 60 patients

## Box 1 Trial objectives

### Primary
► Compare the effectiveness of exenatide once weekly versus placebo on the Movement Disorders Society Unified Parkinson's Disease Rating Scale (MDS-UPDRS) part 3 motor subscore in the 'practically defined OFF medication state' in patients with Parkinson's disease (PD) (change in the MDS-UPDRS part 3 score reflects accumulation of motor deficit and therefore is a measure of PD motor progression).

### Secondary
► Compare differences at 48 and 96 weeks between the exenatide and placebo trial arms in:
  – MDS-UPDRS part 1, 2, 3 and 4 ON medication scores.
  – Timed walk assessment ON and OFF medication.
  – Montreal Cognitive Assessment.
  – Safety and tolerability of exenatide as indicated by changes in vital signs, weight, clinical laboratory measures and adverse events
  – Patient Health Questionnaire.
  – Unified Dyskinesia Rating Scale.
  – Parkinson's Disease 39 item Quality of Life questionnaire.
  – Levodopa equivalent dose change.
  – A 3-day Hauser diary of PD state (Time-On, Off, Non troublesome Dyskinesia, Troublesome dyskinesia, Asleep).
► Compare differences in total values over 96 weeks between the exenatide and placebo trial arms in:
  – Health and social care resource use on the modified Client Service Receipt Inventory.
  – Health and social care costs.
  – Paid and unpaid carer costs.
  – Quality-adjusted life-years calculated using the EQ-5D-5L tariff adjusting for baseline.
► Compare differences between scores at 48 and 96 weeks between the exenatide and placebo trial arms in:
  – MDS-UPDRS part 3 Motor subsection OFF medication score.

### Exploratory
Compare differences between slopes at prespecified periods between exenatide and placebo trial arms for key outcomes to investigate whether exenatide can be considered disease modifying.

EQ-5D-5L, EuroQol- 5 Dimension, 5 Level.

with PD has subsequently been performed.[14] Patients were randomised to self-injection of a long acting form of exenatide, (Bydureon 2 mg) once weekly, or matched placebo for 48 weeks. Detailed assessments every 12 weeks for the duration of the treatment and a further assessment at the 60 weeks time point to explore any lasting effects following washout of the trial medication were performed. Patients receiving exenatide had a mean 3.5 point advantage in their MDS-UPDRS part 3 OFF medication scores compared with patients receiving placebo at the 60 weeks time point. Biological specimens collected from trial participants confirmed changes according to treatment with exenatide in downstream cellular effector pathways.[15]

The current trial objective (box 1) is to confirm or refute whether the previous positive results can be reproduced in a multicentre trial design, including a larger

number of participants evaluated over twice as long a period as previously. An important secondary objective is to explore if positive effects seen after 48 weeks of exenatide exposure remain static or increase in amplitude by the 96 weeks time point. The hypothesis is that exenatide will be associated with reduced MDS-UPDRS part 3 scores at the 96-week time point. The overriding priority for this trial is to provide evidence to support or refute any signal of efficacy of exenatide in PD, and thus provide the justification for rapid further investment in this drug if appropriate. In parallel with this, is the aim to explore whether any biological effect(s) of exenatide, relevant to PD, are purely symptomatic effects as opposed to disease-modifying effects.

## METHODS

This trial protocol was designed using the University College London (UCL) Comprehensive Clinical Trials Unit (CCTU) Protocol template. The trial is sponsored by UCL and coordinated by the CCTU. The protocol was designed to provide information about procedures for entering participants into the trial, and sufficient detail to enable: an understanding of the background, rationale, objectives, trial population, intervention, methods, statistical analyses, ethical considerations, dissemination plans and administration of the trial; replication of key aspects of trial methods and conduct; and appraisal of the trial's scientific and ethical rigour from the time of ethics approval through to dissemination of the results. All stake holders (research team, sponsor, CCTU and oversight committees) were involved in the design and approval of the protocol. A particular emphasis was given to patient input in the trial design. This patient and public involvement (PPI) approach has proven to be of value in other studies[16] and was harnessed to improve the overall study design. A focus group meeting with patients was organised in the protocol design stages to obtain feedback from patients which led to a number of amendments prior to submission, including the maximum overall trial duration of 96 weeks, and the use of OFF-medication assessments. Two PPI representatives will serve on the trial steering committee (TSC) and will continue to provide regular input throughout recruitment. Patients will also be provided access to the trial website and a link to the protocol and patient information sheets (PIS) on request and will be given the opportunity to continue to provide comments and contact researchers to further discuss their input. The INCLUDE guidance[17] is an National Institute for Health Research-led initiative to improve inclusion of under-served groups. The design of the trial is mindful of the value of the steps outlined in this initiative and aims to incorporate its recommendations into overall trial recruitment with the overarching aim of providing better access and quality care to under-served patient groups.

## Patient and public involvement

In the development of this protocol, a formal meeting hosted by the Cure Parkinson's Trust was held with six patients with PD to obtain patient feedback on the overall trial design and logistical aspects of the trial that could potentially impede recruitment and retention. The aims and objectives of the trial were discussed including the importance of distinguishing between symptomatic and disease modifying effects of exenatide. Patient feedback was clear that a 2-year period would be the maximal acceptable duration of self-administration of placebo, therefore, the trial duration was reduced from the original planned 3-year duration to 96 weeks. The use of weekly self-administered injections, and attendance in the off-medication state to assess PD severity was discussed in detail and considered acceptable. The recruitment strategy has used the patient networks of the Cure Parkinson's Trust and Parkinson's UK to increase the awareness of the trial. Patients and patient representatives are included in the TSC. At the end of the study, all participants will be notified of their randomisation allocation and of the main study results. The results will be presented at meetings convened for patient groups and published in open access peer reviewed publications.

## Trial design

This is a simple parallel group multicentre phase 3, double-blind, randomised, placebo-controlled trial which includes a 96-week exposure period. Detailed evaluations of all participants will take place at screening, baseline, 24, 48, 72 and 96 weeks (figure 1). Participants will also attend on a 12 weekly basis to collect supplies of Investigational Medicinal Product (IMP). Participants will be randomly allocated to receive either exenatide extended release 2 mg subcutaneous injection (Bydureon) once weekly for 96 weeks n=100, or exenatide extended release placebo subcutaneous injection once weekly for 96 weeks n=100. In addition, participants will be randomised using a minimisation algorithm (with a random element incorporated) balancing by research site, participants with greater (Hoehn and Yahr stage 2.5) or lesser (Hoehn and Yahr stage 2.0 or less) PD severity (in the ON medication state), and participation in the substudies (remote monitoring, imaging or not participating).

## Participants and recruitment

Patients are eligible for screening if they have a clinical diagnosis of PD. The Queen Square brain bank criteria[18] can be also be used to validate the diagnosis and ensure consistency of diagnosis between sites, however, this is not a formal inclusion criterion. The relevance of a positive family history of PD, or a confirmed genetic basis for an individual's symptoms will be evaluated in the context of other clinical features in determining diagnosis and eligibility. Key inclusion and exclusion criteria are summarised in box 2.

In a post hoc analysis of the Exenatide PD phase 2 trial, younger patients with shorter disease duration had the

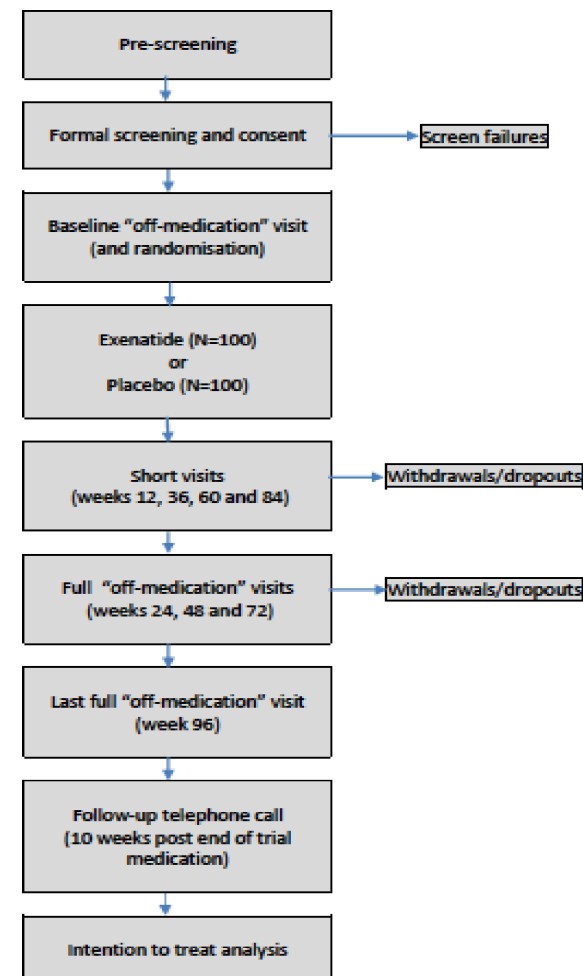

**Figure 1** Outline of trial design.

best outcomes.[19] While aware of this, we feel it is important to collect evidence to determine whether exenatide has beneficial effects on cognition and axial features of PD, and thus took the decision to keep the inclusion criteria broad to improve our chances of detecting effects on these other outcomes and also ensuring that the results will be applicable to the broadest population of PD patients.

Participants will typically be recruited through specialist movement disorders clinics at trial sites. The trial will be advertised online by the Parkinson's UK website, the Cure Parkinson's Trust and the NIHR Clinical Research Network websites and will be registered on Clinical-Trials.gov and the ISRCTN registry. Trial advertisements will direct participants to contact teams in order to be provided with a PIS and a reply slip to confirm ongoing interest and to organise a prescreening telephone call to discuss eligibility and suitability for the study. It is anticipated that recruitment will be completed from six UK sites (National Hospital for Neurology and Neurosurgery (Queen Square, London), King's College Hospital National Health Service (NHS) Foundation Trust (London), Oxford University Hospitals NHS Foundation Trust (Oxford), Derriford University Hospital (Plymouth), Salford Royal Hospital (Manchester) and Western General Hospital & Royal Infirmary of Edinburgh

---

**Box 2    Key inclusion and exclusion criteria for study**

**Key inclusion criteria**

► Diagnosis of Parkinson's disease (PD) based on review of the participant's clinical history, examination findings and response to PD medications. The Queen Square brain bank criteria[18] can be also be used to assist the diagnosis, however this is not a formal inclusion criterion. The relevance of a positive family history of PD, or a confirmed genetic basis for an individual's symptoms will be evaluated in the context of other clinical features in determining diagnosis and eligibility.

► Hoehn and Yahr stage ≤2.5 in the ON medication state. This implies that all patients will be mobile without assistance during their best 'ON' medication periods.

► Between 25 and 80 years of age.

► On dopaminergic treatment for at least 4 weeks before enrolment. All participants must have had previous or ongoing exposure to dopaminergic treatment either as L-dopa or a dopamine agonist. If L-dopa has been stopped due to side effects or lack of response, the local PI should further confirm that the participant has clinical symptoms and signs and/or radiological investigations consistent with a diagnosis of PD.

► Ability to self-administer, or to arrange carer administration of trial medication.

► Documented informed consent to participate.

**Key exclusion criteria**

► Diagnosis or suspicion of other cause for Parkinsonism. Patients with clinical features indicating a diagnosis of progressive supranuclear palsy, multiple system atrophy, drug-induced Parkinsonism, dystonic tremor or essential tremor will not be recruited.

► Patients unable to attend the clinic visits in the practically defined OFF medication state.

► Body mass index <18.5. (Exenatide is known to cause weight loss therefore individuals that may not tolerate further weight loss will not be recruited).

► Known abnormality on CT or MRI brain imaging considered likely to compromise compliance with trial protocol.

► Significant cognitive impairment defined by a score <21 on the Montreal Cognitive Assessment.

► Concurrent severe depression defined by a score ≥16 on the Patient Health Questionnaire.

► Prior intracerebral surgical intervention for PD. Patients who have previously undergone deep brain stimulation, intracerebral administration of growth factors, gene therapy or cell therapies will not be eligible.

► Previous participation in one of the following PD trials (Biogen SPARK trial, Prothena Pasadena trial, Sanofi Genzyme MOVES-PD trial, UDCA-PD UP Study or any other trial still considered to involve a potentially PD modifying agent). In the event of any uncertainty, the chief investigator will discuss the relevance of exposure to any other specific trials/experimental agents with the local Principal Investigator before recruitment eligibility is confirmed.

► Participation in another clinical trial of a device, drug or surgical treatment within the last 30 days.

► Previous exposure to exenatide.

► Impaired renal function with creatinine clearance <50 mL/min.

► History of pancreatitis. Screening serum amylase value must fall within laboratory normal range±50%.

► Type 1 or type 2 diabetes mellitus.

► Severe gastrointestinal disease (eg, gastroparesis).

Continued

---

## Box 2 Continued

► Hyperlipidaemia. A lipid profile will be tested at the screening visit. Cholesterol or triglyceride levels greater than 2 × the upper limit of normal will raise suspicion of a familial or acquired hyperlipidaemia and will prompt referral to a relevant specialist for investigation and treatment.

► History or family history of medullary thyroid cancer. Undiagnosed neck lump, hoarse voice or difficulty swallowing (not attributable to PD diagnosis).

► Multiple endocrine neoplasia 2 syndrome.

► Hypersensitivity to any of exenatide's excipients.

► Females that are pregnant or breast feeding. There are no safety data regarding exenatide use in pregnancy.

► Women of childbearing potential who are unwilling or unable to use an acceptable method to avoid pregnancy for the entire trial period and up to 3 months after the last dose of trial medication. Female participants who are able to become pregnant (defined as women of childbearing potential) will undergo a pregnancy test prior to randomisation and will be asked at each visit to confirm regular use of an effective method of contraception

► Participants who lack the capacity to give informed consent.

► Any medical or psychiatric condition or previous conventional/experimental treatment which in the investigator's opinion compromises the potential participant's ability to participate.

UDCA, Ursodeoxycholic acid.

(Edinburgh). Recruitment rates will be carefully monitored throughout the trial to inform on the total number of sites required to ensure final recruitment milestones will be reached. All patient assessments will be performed at hospitals in the UK, after a site initiation visit has been performed. The trial began recruitment on 20 January 2020 and will aim to complete all assessments by 30 September 2023.

## Outcomes
### Safety monitoring

Safety and tolerability of exenatide as indicated by changes in vital signs, weight, clinical laboratory measures and adverse events (AEs) will be recorded and monitored throughout. Each patient will have their pulse, blood pressure and weight documented at screening and at each follow-up visit. Exenatide is known to cause weight loss. Participants' height will be recorded at screening to enable calculation of body mass index. At each visit, participants are asked to report any AEs that have occurred since the previous visit. AEs may also be detected by the study team reviewing the patient or through notification by the participant's primary care physician. All AEs will be assessed by a study doctor for their severity, likely relationship to study drug and required action by a study doctor not involved in the blinded assessment of the patient. All SAEs will be recorded and reported to the sponsor regardless of relation to trial treatment. Any suspected unexpected serious adverse reactions will be reported to the sponsor immediately to allow facilitation of unblinding as necessary. All AEs reported will

be reviewed by the trial management group (TMG), trial steering group and monitored by an Independent Data Monitoring Committee (IDMC). Unblinding requests from other clinicians responsible for a patient's care will be handled by the principal investigator (PI) at each site. The PI at each site may also choose to unblind a participant in response to reported AEs as they are reported, if judged to be clinically necessary.

### Primary outcome

The MDS-UPDRS part 3 motor OFF medication score is a widely accepted measure of the motor disability of PD. The scale is performed in the ON medication state and in the practically defined OFF medication state. This is defined as the score obtained in a patient who has withheld all short acting conventional PD medications for at least 8 hours and all long acting conventional PD medications for at least 36 hours. Comparison of MDS-UPDRS part 3 motor subscore in the practically defined OFF medication state at 96 weeks between participants according to treatment allocation and adjusted for baseline will be the primary outcome. The scores for these assessments will be collected and recorded by trained clinical trial personnel (if possible, the same person will rate these assessments at each site to minimise inter-rater variability). With consent, these assessments will be video recorded as part of an MDS-UPDRS automated scoring sub study though the availability of these videos will also enable repeated independent scoring to be performed if there are concerns raised about data quality from a specific site/rater.

### Secondary outcomes

Comparisons at 48 and 96 weeks between participants according to treatment allocation will also be performed for each of the secondary outcomes listed below.

### MDS-UPDRS part three motor score in the practically defined OFF medication state

Whereas the analysis of the 96-week scores according to randomisation group will represent the primary outcome for this trial, differences emerging at 48 weeks and also the difference between scores at 48 weeks and 96 weeks will be important secondary outcomes.

### MDS-UPDRS part 1, 2, 3 and 4 on medication scores

Part 3 of the MDS-UPDRS as well as the other elements (part 1, 2 and 4) of the scale will also be evaluated in the presence of conventional PD medication (ON state) to evaluate any change in some of the NMS of PD, activities of daily living and the complications of chronic PD treatment.

### Montreal Cognitive assessment

This scale is a validated global measure of cognitive ability. This will be assessed in the ON medication state.

### Timed tests

Participants will be asked to perform a Sit-stand-walk timed test in both the OFF medication and ON medication state. The timed Sit-stand-walk test will incorporate

time taken from seated position to stand and walk 10 metres, turn and return to original seated position.

### Unified Dyskinesia Rating Scale
This is considered to be the most useful and objective way of quantifying dyskinesia severity. This will be assessed in the ON medication state.

### Patient Health Questionnaire-9
This scale allows for self-quantification of depression severity. This will be assessed in the ON medication state.

### Non-Motor Symptom Scale
This validated scale is a tool to collect data on the frequency and severity of 30 NMS sometimes experienced by PD patients. This will be assessed in the ON medication state.

### The Parkinson's Disease Questionnaire
This is the standard disease specific measure of quality of life in PD comprising 39 questions. It has been extensively validated in previous studies.

### Levodopa equivalent dose
To facilitate comparisons between patients taking different regimes of conventional PD medications, a set of conversion factors have been used to convert each of the commonly used PD medications to an LED of each of their medications can then be summed for interpatient/intergroup comparisons.[20]

### EQ-5D-5L
This is a simple, 5 question form and visual analogue scale that allows calculation of quality-adjusted life-years to enable health economic analyses to be performed.

### The Client Service Receipt Inventory
Health and social care resource use. Self-completed healthcare, social care and paid/unpaid carer resource use questionnaire asking about primary and secondary care resource use relevant to Parkinson's and impact on carers in the past 6 months.

### Three-day Hauser diary
A 3-day Hauser diary of PD state (time-on, off, troublesome dyskinesia, non-troublesome dyskinesia, asleep). Diary data allow quantification of the amount of time during a 3-day period that patients spend in the varying states of movement ability.

## Ancillary studies
There are four optional substudies linked to the main trial:

1. Genetics substudy: To try to identify genetic markers that may be associated with subtypes of PD or variation in treatment responsiveness.
2. Cerebrospinal Fluid (CSF) substudy: To determine whether any CSF changes associated with PD are influenced by exposure to exenatide. These may include alpha synuclein monomers or oligomers, neuroinflammatory markers, and exosomal contents.
3. Remote Monitoring of PD Symptoms substudy: To help determine whether measurement of PD symptoms using digital technology may be a more sensitive measure of change with active drug vs placebo compared with the MDS UPDRS 3 in the OFF and ON medication states. This will form two separate measurements comprising (1) home-based smartphone and (2) real-world gait/walking activity monitoring.[21–23] This aims to generate precision data, providing person-specific distributions of outcomes and may be able to better delineate baseline clinical features.
4. DaTSCAN (Imaging substudy): To determine if change in dopamine transporter availability in the caudate and putaminal nuclei as measured by quantitative DaTSCAN signal is influenced by exposure to exenatide compared with placebo.

## Visits
The overall progression of assessments are summarised in figure 1. While we expect to undertake all assessments in respective clinical units, provision has been made in line with INCLUDE guidance for the possibility of home visit assessments to be performed when patient specific situations (eg, inability to travel due to coronavirus restrictions, worsening 'OFF' state over progression of trial) necessitate this. We hope that this provision will aid overall trial retention while enhancing recruitment of patients from typically less well represented demographics (eg, rural geographical regions, patients lacking private travel facilities).

### Screening visit
Written informed consent to enter and be randomised into the trial will be obtained from participants, after explanation of the aims, methods, benefits and potential hazards of the trial and before any trial-specific procedures are performed or any blood is taken for the trial. Patients will be screened using the history of their PD, supported by any available clinical correspondence according to usual standard of care.

The collection of the following scales will evaluate patient eligibility: Montreal Cognitive Assessment (MoCA), PHQ-9, as well as blood tests (full blood count, urea and electrolytes, creatinine, liver function tests, Haemoglobin A1c, C-peptide, coagulation, serum amylase, thyroid function tests, blood glucose, insulin and lipid profile, and a pregnancy test for women of childbearing potential). Tests can be repeated between screening and baseline visits, if required to confirm eligibility. Abnormalities detected that warrant further management for example, newly diagnosed diabetes will be referred for appropriate medical evaluation.

Patients recruited to the DATSCAN substudy will have imaging performing prior to their baseline visit.

### Baseline visit and randomisation
Previously defined primary and secondary outcome measures will be performed in the 'ON' and 'OFF' states

as outlined below. Patients' LED will be noted. Randomisation to either exenatide or placebo will be administered using a centralised, web-based system (www.sealedenvelope.com). All assessments related to sub-studies will also be performed prior to trial medication administration.

## Assessment procedures

After the screening visit, the named site clinical staff member will call the participant to remind them of the need to stop taking their regular PD medication prior to their next trial visit and to attend in a fasted state (prior to visits 2, 4, 6, 8, 10). The MDS-UPDRS part 3 and Timed Walk assessments will be initially performed in the OFF state. This assessment in both the 'OFF' and 'ON' states will be performed with video recording to facilitate the possibility of a re-review if necessary. Remote monitoring assessments will be conducted at this point at selected sites in patients consenting to participate in this substudy. While waiting for medications to work, participants will self-complete the MDS-UPDRS parts 1, 2 and 4, Parkinson's Disease Questionnaire-39, EQ-5D-5L and Client Service Receipt Inventory. The MDS-UPDRS part 3 and Timed Walk assessments will be repeated 1 hour after the participant has taken their routine medications—the ON medication state. After completion of the MDS-UPDRS and Timed Walk assessments in the ON medication state, each participant will be assessed using the MoCA, NMS scale, Unified Dyskinesia Rating Scale and PHQ-9. This will occur in alternate postrandomisation assessments (at visits 2, 4, 6, 8, 10). At selected centres participants in the CSF substudy will have a CSF sample taken via lumbar puncture. Ten weeks after the last trial medication administration, a staff member will call the participant to collect details of any AEs that have occurred after the participant stops taking the trial medication. Participants will complete the 3-day Hauser Diary prior to visits 2, 6 and 10 and return the diary back to the research team at the respective study visits. At each of the visits 2, 4, 6, 8 and 10, a blood sample will be collected and processed for storage for future analysis.

The DaTSCAN imaging substudy will be performed at the UCLH site on all consenting substudy participants; scans will be performed prior to visit 2 and after visit 10.

The option for performing a remote assessment will be provided to patients for safety monitoring visits in view of the coronavirus pandemic.

## Intervention

Each dose of exenatide 2 mg (powder and solvent for prolonged release, suspension for injection, prefilled pen) is supplied as a single use injection pen for subcutaneous administration by the patient on a weekly basis. The placebo (inactive powder and solvent for prolonged release, suspension for injection, prefilled pen) is supplied as an identical injection pen for subcutaneous administration by the patient on a weekly basis. The trial medication will be refrigerated and stored at 2°C–8°C. Both exenatide and placebo will be supplied by AstraZeneca

as unlabelled prefilled pens in bulk and in accordance with Good Manufacturing Practice (GMP). Labelling, packaging and release of packed trial medications will be managed by the Sponsor's contracted company following GMP. The labels will be prepared in accordance with GMP Annex 13 (online supplemental material 1) requirements for labelling and local regulatory guidelines. The trial medications will be released ahead of trial use.

Site trial staff will be trained on the use of exenatide using an online teaching video, accompanying product literature and the investigator's brochure. Patients will be taught how to perform the subcutaneous injections by the clinical trial team (TT) using the online video, demonstration packs and written literature. They will be told about common adverse reactions previously reported, for example, nausea, vomiting, diarrhoea and weight loss by the clinical TT, and will be advised on the processes for safety reporting. In the event that exenatide injections will be administered by caregivers (eg, spouse), their willingness to perform this will be documented and they will be trained using the online teaching video. It will also be ascertained that the caregiver either lives with the PD patient or confirms their willingness to meet with the PD participant on a weekly basis to administer the injections for the 96-week period of the trial.

Patients who meet eligibility criteria at the screening visit will be randomly assigned to receive 96 weeks of double-blind treatment with either exenatide or placebo (2 mg once weekly) in a 1:1 ratio. The first dose will be administered by the patient in clinic following injection training and subsequent injections will be at home. Injections will be self-administered by the participants, or administered by their carer, into the participants' abdomen, arm, thigh or buttocks every 7 days. Participants will be provided with a link to the injection pen training video and an Research Ethics Committee approved injection administration training sheet.

## Sample size

The sample size is based on the detectable effect size (primary outcome is the MDS-UPDRS motor subsection in the OFF medication state) for a two-arm (exenatide vs placebo) parallel-group trial design. The calculations assume a common SD of 13.5, and a correlation of 0.70 between baseline and follow-up MDS-UPDRS measurements. These estimates are reasonable based on data from the previous exenatide-PD trial.[14] On this basis, 160 evaluable participants divided equally between the two groups is sufficient to detect a difference of 5.0 MDS-UPDRS part 3 points in the OFF medication state between the two groups adjusting for baseline MDS-UPDRS part 3 OFF scores, with 90% power and at a significance level of 0.05. Assuming 20% attrition (withdrawal/lost to follow-up), 200 participants will be recruited. Participants who withdraw from the trial will not be replaced. Participants who withdraw from trial treatment should remain in the trial for the purpose of follow-up and data analysis. This effect size is a reasonable expectation based on the previously

collected pilot data and would represent a clear demonstration of the efficacy of exenatide on the motor severity of PD.

It is also anticipated that the difference in scores in the ON medication state will be greater at 96 weeks than at the earlier time points. The expected rate of change in PD severity in the first 5 years after PD diagnosis in the ON medication state is 1 MDS-UPDRS part 3 point per year. A predicted advantage of 2 points in ON scores over 96 weeks would thus equate to an advantage in the rate of disease progression above and beyond that achievable with conventional dopaminergic medication and would be a further clear signal that continued use of exenatide is consistent with not only long-term disease modifying effects, but even demonstration of a small change of 2.5 points in the MDS-UPDRS motor score would constitute a clinically important difference[24] and potentially an advantage in day to day functional impairment and overall improvement in quality of life in the short term.

## Statistical analysis

A full statistical analysis plan will be written and approved by the TSC prior to database lock. All analyses will be undertaken according to a modified intention-to-treat principle in accordance with the randomised intervention. The threshold for the analysis population will be participants who complete 12 weeks on treatment and for whom outcomes are available.

Primary outcome analysis will evaluate the impact of treatment allocation (exenatide or placebo) on the difference between MDS UPDRS part 3 OFF medication scores at 96 weeks follow-up adjusting for baseline. The analysis will use a mixed-model approach incorporating information from all follow-up visits that adjusts for baseline Hoehn and Yahr status and the baseline raw value of each outcome measurement. Site will be included as a random effect to account for variability in outcomes between sites, and a random patient/subject effect will accommodate the correlation between repeated outcome measures on the same patient. A significance level of 5% will be used to judge significance for the primary outcome measure.

A planned secondary analysis will compare the difference in MDS-UPDRS part 3 OFF medication scores according to randomisation allocation at 96 weeks, with the scores at 48 weeks. An increase in the advantage at 96 weeks compared with 48 weeks would be evidence that the active drug was slowing down disease deterioration rather than having symptomatic effects only. This could translate to a major population advantage in terms of reduction of morbidity and mortality.

Analyses of the remaining secondary/exploratory outcomes will be undertaken similarly for the difference between groups according to treatment allocation at 48 and 96 weeks follow-up adjusting for baseline values of each outcome, and confounding factors such as LED differences between groups.

Further exploratory analyses will consider whether exenatide can be thought of as disease modifying by comparing slopes between groups at prespecified periods.

A sensitivity (per-protocol) analysis will be performed for the primary outcome measure and will only include those participants who completed the trial in accordance with the approved protocol.

Results on the primary efficacy outcome will be presented by stratum, according to Hoehn and Yahr stage (≤2.0 vs 2.5), and an interaction between Hoehn and Yahr and treatment will be added to the primary analysis model to investigate whether the effect of treatment differs according to the Hoehn and Yahr stage.

All analyses will be performed by the designated trial statistician.

## Data management

Data will be entered in the Exenatide-PD3 database by delegated staff at participating sites and members of the Exenatide-PD3 TT at CCTU. Participants will be given a unique trial PIN (Exnnn). Data will be entered under the Personal Identification Number onto the central database (InferMed's MACRO stored on the servers based at UCL). The database will be password protected and only accessible to members of the Exenatide-PD3 TT and external regulators if requested. Video recordings of the MDS-UPDRS will be uploaded onto a secure cloud held by Machine Medicines Technologies (MMT) and used for quality control purposes. Appropriate contractual agreements covering data protection are in place with MMT. All data storage will adhere to GDPR and the Data Protection Act 2018.

An IDMC will be convened including at least three individuals independent from the TT and sponsor who have experience in the conduct of clinical trials for PD. The IDMC will review the trial results and make a recommendation to the TSC regarding continuation/stopping of the trial based on safety data. A statistician independent of the Exenatide-PD3 TT at CCTU will generate summaries of accumulating trial data for the IDMC to review.

UCL is the trial sponsor and has delegated the duties as sponsor to CCTU via a signed letter of delegation. The trial sponsor will take on responsibility for securing the arrangements to initiate, manage and finance the trial. Trial oversight is intended to preserve the integrity of the trial by independently verifying processes and prompting corrective action where necessary. In multicentre trials this oversight is considered and described both overall and for each recruiting centre by exploring the trial dataset or performing site visits. The TT will assist with developing the design, coordination and day-to-day operational issues in the management of the trial, including budget management. The TMG will assist with developing the design, co-ordination and strategic management of the trial. The independent TSC is the independent group responsible for oversight of the trial in order to safeguard the interests of trial participants. The TSC will provide advice to the chief investigator, CCTU, the funder and

sponsor on all aspects of the trial through its independent Chair. The IDMC is the only oversight body that has access to unblinded accumulating comparative data. The IDMC will be responsible for safeguarding the interests of trial participants, monitoring the accumulating data and making recommendations to the TSC on whether the trial should continue as planned. The membership, frequency of meetings, activity (including trial conduct and data review) and authority of each committee will be covered in their respective terms of reference.

## ETHICS AND DISSEMINATION

The trial protocol, all informed consent forms and any material to be given to the prospective participant have received REC (initial date of approval 15/10/2019, REC reference no.19/SC/0447), and other regulatory approvals (EudraCT 2018-003028-35). Further, the trial was registered in clinicatrials.gov NCT004232969 and in ISRCTN (reference 14552789). Subsequent amendments to these documents will be submitted for further approval. The same/amended documents will be submitted for additional local permissions at each clinical site.

This is a Clinical Trial of an IMP as defined by the EU Directive 2001/20/EC. Therefore, a clinical trial authorisation is required in the UK and the trial protocol will therefore be submitted to the UK regulatory authority (MHRA). The progress of the trial, safety issues and reports, including expedited reporting will be reported to the MHRA as required. The protocol, PIS and informed consent forms on local headed paper, the REC/HRA and MHRA approvals, schedules of funding and activity (and other trial documentation as needed) have been submitted to the relevant NHS Trust Research & Development department of each participating site or to other local departments for approval.

Participants will be provided with a PIS and given time to read it fully. Following a discussion with a medical qualified investigator or suitable trained and authorised delegate, any questions will be satisfactorily answered and if the participant is willing to participate, written informed consent will be obtained (online supplemental material 2). During the consent process, it will be emphasised that the participant is free to refuse to participate in all or any aspect of the trial, at any time and for any reason, without affecting their treatment. The risk/benefit profile of the trial will be regularly monitored. Consent will be resought if new information becomes available that affects the participant's consent in any way.

The rights of the participant to refuse to participate in the trial without giving a reason will be respected and after the participant has entered the trial, the clinician remains free to give alternative treatment to that specified in the protocol, at any stage. The participant remains free to change their mind at any time about the protocol treatment and follow-up without giving a reason and without prejudicing their further treatment. All participants will be made aware of the known adverse reactions.

## DISCUSSION

A parallel group design with a washout period has been used previously in the evaluation of potential neuroprotective agents.[25 26] and this was chosen as the design for the previous phase 2 trial. This design is subject to possible long duration symptomatic effects and a lengthy washout period potentially impacts on patient retention and cannot necessarily distinguish a true neuroprotective effect from a symptomatic effect (in view of preservation of healthy behaviours with long term impacts such as exercise).[27] An alternative approach which we have adopted here is a 'long-term simple' design, with longer-term follow-up to look for a cumulative advantage emerging with prolonged treatment exposure, given the natural history of PD being that of progressive accumulation of motor and non-motor disability.[28] This design helps build on the previous successful clinical trials of exenatide which have introduced a novel, cost effective way of evaluating the potential for disease modifying drugs in PD by recruiting patients already in receipt of conventional dopaminergic treatment, rather than restricting recruitment to incident cases yet to receive dopaminergic treatment. Using this approach, we have successfully demonstrated the potential for rapid recruitment, and improved retention of participants enabling more complete follow-up, and a statistically significant advantage in motor scores in people randomised to exenatide over a 48-week period of treatment exposure. We have considered that an exposure period of 96 weeks would allow exploration of long-term effects of exenatide exposure, while being the maximum period that participants would be willing to accept being allocated placebo. Furthermore, this will provide the opportunity to evaluate whether the 48-week data previously published can be replicated and whether effects at 96 weeks are similar to or greater than those seen at 48 weeks and other earlier time points.

Many trials have attempted to evaluate the potential for disease modification using drugs with broad mechanisms of action. The majority have either failed to demonstrate clinical efficacy or provided inconclusive results. Some pertinent reasons for this are a failure of the investigated agent to reach and engage its target and lack of objective measures of true clinical disease progression. Clinical endpoints such as the MDS-UPDRS scale are necessary to ultimately confirm relevance; however, these scales lack sensitivity for capturing disease modification unless very long term follow-up data are collected. While our approach of following patients for 2 years in this trial will partially mitigate some of this, the addition and to an extent validation of more detailed approaches through a number of sub studies (imaging, CSF analysis for target engagement and drug levels, and device assisted measurements of real-life motor function) could ultimately provide more holistic and definitive metrics for determining if exenatide does in fact deliver disease modification in PD. This more comprehensive approach to approaching assessments and the consistent signal of

benefit noted in two earlier trials provide grounds for optimism that the primary outcome will be achievable. The study opened to recruitment in January 2020 and we expect completion of study analysis by Q3 2024.

**Author affiliations**
[1] Department of Clinical and Movement Neurosciences, UCL Queen Square Institute of Neurology, London, UK
[2] National Hospital for Neurology and Neurosurgery, London, UK
[3] The Comprehensive Clinical Trials Unit, UCL, London, UK
[4] Surrey Clinical Trials Unit, University of Surrey, Guildford, UK
[5] Department of Clinical & Experimental Medicine, University of Surrey, Guildford, UK
[6] Applied Parkinson's Research Group, University of Plymouth, Plymouth, UK
[7] University Hospitals Plymouth NHS Trust, Plymouth, UK
[8] Nuffield Department of Clinical Neurosciences, University of Oxford, Oxford, UK
[9] Oxford Parkinson's Disease Centre, University of Oxford, Oxford, UK
[10] Department of Clinical Neurology, Oxford University Hospitals NHS Foundation Trust, Oxford, UK
[11] Department of Neurology and Neurosurgery, University of Manchester, Greater Manchester, UK
[12] Western General Hospital, NHS Lothian, Edinburgh, UK
[13] University of Edinburgh, Edinburgh, UK
[14] Parkinson's Foundation International Centre of Excellence, King\'s College London, London, UK
[15] Translational & Clinical Research Institute, Newcastle University, Newcastle upon Tyne, UK
[16] Newcastle Upon Tyne NHS Foundation Trust, Newcastle, UK
[17] London, UK
[18] Department of Nuclear Medicine, University College London Hopsitals NHS Trust, London, UK
[19] Research Dept of Primary Care and Population Health, University College London, London, UK
[20] Leonard Wolfson Experimental Neurology Centre, National Hospital for Neurology & Neurosurgery, London, UK
[21] University College London, London, UK

**Acknowledgements** We are grateful to all patients contributing to the design of this study. We thank the Cure Parkinson's Trust and Parkinson's UK for their assistance in the setup and increasing the awareness of this study.

**Contributors** The study concept and design was conceived by TF. NV and CG will conduct screening and data collection. GA, MC, KM, KC, HM, SH, PL, DA, CBC, MTH, MS, GWD, RC, CL, SDD, AJY, LR, RG, JD, RH and VL assisted on drafting the protocol. Analysis will be performed by SS, and data entry by AK. NV prepared the first draft of the manuscript. RG served as a patient adviser for the design of the study protocol. All authors provided edits and revised the manuscript for intellectual content.

**Funding** This work was supported by NIHR grant number 16/167/19. The Cure Parkinson's Trust funded the DATSan Imaging substudy grant number BY-TF021 and the Remote Monitoring substudy grant number MH011.

**Competing interests** None declared.

**Patient consent for publication** Not required.

**Provenance and peer review** Not commissioned; externally peer reviewed.

**ORCID iDs**
Nirosen Vijiaratnam http://orcid.org/0000-0002-9671-0212
Simon Skene http://orcid.org/0000-0002-7828-3122
Camille B Carroll http://orcid.org/0000-0001-7472-953X
Thomas Foltynie http://orcid.org/0000-0003-0752-1813

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
