## [Reviewer comments · BMJ Open]

ARTICLE DETAILS

TITLE (PROVISIONAL)	Exenatide once weekly over 2 years as a potential disease modifying treatment for Parkinson's disease: protocol for a multi-centre, randomised, double blind, parallel group, placebo controlled, Phase 3 trial, The 'Exenatide-PD3' study
AUTHORS	Vijiaratnam, Nirosen; Girges, Christine; Auld, Grace; Chau, Marisa; Maclagan, Kate; King, Alexa; Skene, Simon; Chowdhury, Kashfia; Hibbert, Steve; Morris, Huw; Limousin, Patricia; Athauda, Dilan; Carroll, Camille; Hu, Michele; Silverdale, Monty; Duncan, Gordon W; Chaudhuri, Ray; Lo, Christine; Del Din, Silvia; Yarnall, Alison J; Rochester, Lynn; Gibson, Rachel; Dickson, John; Hunter, Rachael; Libri, Vincenzo; Foltynie, Thomas

VERSION 1 – REVIEW

REVIEWER	Banks, William Geriatrics Research Education and Clinical Center, Veterans Affairs Puget Sound Health Care System
REVIEW RETURNED	02-Feb-2021

GENERAL COMMENTS	1) Statement in Exclusion that any newly diagnosed diabetics will be referred for appropriate medical evaluation/care. 2) Ancillary Studies: Consider including CSF/serum albumin ratios as some evidence it is elevated in later stages of PD. 3) Though the authors make a case for not including a washout period, I still think that they should include one as was done in the prior open label study. There is a substantial literature on the benefits of doing this (Ploeger BA and Holford NHG, Pharmaceut Statist 8:225-238, 2009). 4) I would use a formal set of criteria to clinically define PD – the authors state that the “Queen Square brain bank criteria[18] can be also be used to assist in the diagnosis, however, this is not a formal inclusion criterion.” These criteria (also known as the UKBB Criteria) are what we, NINDS, and many other groups use for prevalent PD cohorts. But rather than “assisting” in the diagnosis, these should be used to define the clinical diagnosis in an effort to decrease clinical heterogeneity (since multiple sites are involved). 5) If one just plans to use a single brief and global measure of cognition in PD, then I would use the MoCA (it's more widely used than any other global battery in PD). However, my problem with the study design is that cognition is an IMPORTANT secondary outcome – so why just use the MoCA? I would strongly encourage the authors to use a more comprehensive battery like the one that PPMI is using (https://www.michaeljfox.org/ppmi-clinical-study). These authors know all about PPMI – the battery takes more time but it's not unreasonable – I would do it at baseline, 48 weeks, and
---

	96 weeks. This will be much more sensitive to protective effects over a 96 week trial period. 6) I'm glad that they are getting additional information using remote monitoring and DaTSCAN - that's a strength.
--	--

REVIEWER	Glotfelty, Elliot NIH, Drug Design and Development
REVIEW RETURNED	05-Feb-2021

GENERAL COMMENTS	The current protocol for a Phase 3 clinical trial using Exenatide for the treatment of Parkinson's Disease (PD) is an important advancement for the repurposing of an already approved T2DM therapy. Building on the success of a previous 48-week clinical trial with Exenatide in PD patients, the authors of the current protocol aim to repeat these positive results in the primary outcome (comparison of MDS-UPDRS part 3 motor sub-score in the practically defined OFF medication state) with additional secondary objectives added. One major modification to the previous clinical trials of Exenatide and PD is that it will occur at multiple in centers across the UK. This is important for transparent and reliable collection of data and will lend further credence to the data collected. The protocol is well designed, taking into account vast input from a variety of stakeholders, including most importantly, PD patients. Below are a few questions, comments, points of clarification, and suggested minor revisions: -Exenatide is capitalized in some cases and in other cases not. Stay consistent throughout the protocol. -In Box 1, please clarify "levodopa equivalent doses"--- later in the protocol, it is mentioned that the LED is used to account for different dopamine replacement regimens. Perhaps stating "levodopa equivalent dose analysis" would be clearer for Box 1. Explanation of LED on page 13 is worded awkwardly. Please have a look and reword. Line 36 Page 9: "In the development of this protocol, a formal meeting was help with 6 patients with PD, hosted by the Cure Parkinson's Trust." Please clarify the role of these 6 patients. Wording needs adjusted. Lin3 Page 10: Clarify what "IMP" is. It is later clarified on page 16. Line 30 Page 10: Remove parentheses BOX 2:
---

	Inclusion criteria point 1 is redundant and a copy/paste from the first paragraph of participants and recruitment. Perhaps shorten to the first sentence without mention of the Queen Square bank criteria which is mentioned in the text. Comment: Based on your previous post-hoc analysis of the data from the Phase 2 study (Athauda et al., 2019), it was found that a subset of patients were better responders to the exenatide treatment. Is any special effort being made to recruit patients based on these findings (earlier disease progression, age, etc.)? The age inclusion for the current study was increased to 80 (from 75) and the Hoehn and Yahr stage ≤ 2.5 inclusion criteria remains the same. I would be interested in any comment on this analysis and whether this analysis affected or had no effect on decisions for inclusion criteria. On page 17 there is mention of the comparative analysis between patients with lower and higher Hoehn and Yahr stage scores. Citation of (Athauda et al., 2019) would be nice here and provide some context. Line 8 Page 16: Missing a period after citation 14. Line 38 Page 16: Please clarify if the 12 -week participation is the threshold for participant data inclusion. Is this a minimum participation for inclusion in analyses?
--	--

VERSION 1 – AUTHOR RESPONSE

Reviewer: 1

Comment:

Statement in Exclusion that any newly diagnosed diabetics will be referred for appropriate medical evaluation/care.

Response:

A statement has been included in the screening visit section to suggest that all abnormalities detected that warrant further medical input will be referred for appropriate medical evaluation.

Comment:

Ancillary Studies: Consider including CSF/serum albumin ratios as some evidence it is elevated in later stages of PD.

Response:

We thank the reviewer for their valid suggestion for future CSF analysis. This will certainly be deliberated upon as part of a comprehensive future plan for analysis of CSF.

Comment:

Though the authors make a case for not including a washout period, I still think that they should include one as was done in the prior open label study. There is a substantial literature on the benefits of doing this (Ploeger BA and Holford NHG, Pharmaceut Statist 8:225-238, 2009).

Response:

We acknowledge the value of washout periods in clinical trials exploring disease modification. As the reviewer very clearly outlines this was used for further exploration of exenatide's potential benefits in our initial open label study and has been utilized by several other studies exploring disease modification. We have however, opted to perform a long-term follow-up study for several reasons outlined and as we have well commenced the trial will not be able to modify our approach. We do however thank the reviewer for stressing this point.

Comment:

I would use a formal set of criteria to clinically define PD – the authors state that the “Queen Square brain bank criteria[18] can be also be used to assist in the diagnosis, however, this is not a formal inclusion criterion.” These criteria (also known as the UKBB Criteria) are what we, NINDS, and many other groups use for prevalent PD cohorts. But rather than “assisting” in the diagnosis, these should be used to define the clinical diagnosis in an effort to decrease clinical heterogeneity (since multiple sites are involved).

Response:

We acknowledge the challenge of disease heterogeneity in PD and its significant impact on outcomes. While we have not mandated that all patients recruited must fulfil Queen Square brain bank criteria, it is strongly recommended as a guide for the diagnosis of PD and enrolment. All patients recruited into the trial attend specialist movement disorders centres that tend to provide a high degree of diagnostic certainty and who adhere to the criteria when making a diagnosis of PD. The formal Inclusion criteria are approved by our Research Ethics Committee and applied to patients already recruited, and while we accept the sentiment of the reviewer's comment, we cannot change the exact wording of this criterion.

Comment:

If one just plans to use a single brief and global measure of cognition in PD, then I would use the MoCA (it's more widely used than any other global battery in PD). However, my problem with the study design is that cognition is an IMPORTANT secondary outcome – so why just use the MoCA? I would strongly encourage the authors to use a more comprehensive battery like the one that PPMI is using (<https://www.michaeljfox.org/ppmi-clinical-study>). These authors know all about PPMI – the battery takes more time but it's not unreasonable – I would do it at baseline, 48 weeks, and 96 weeks. This will be much more sensitive to protective effects over a 96 week trial period.

Response:

We agree with the author on the value of detailed cognitive testing and the limitations of the MoCA. This trial is however a Phase 3 evaluation and was designed to ensure recruitment and assessments could all be performed quickly with minimal burden on trial centres. We do feel that the MoCA is sufficiently sensitive to detect any important cognitive benefits of exenatide over time, especially given the 96 week follow up period. As the trial has now well commenced with recruitment well underway, we are unable to do additional comprehensive cognitive testing.

Comment:

I'm glad that they are getting additional information using remote monitoring and DaTSCAN - that's a strength.

Response:

We appreciate the reviewers enthusiasm for our DATscan sub study. This was adopted based on promising signal trends noted in our phase 2 randomised trial and would certainly lend additional support to demonstrating disease modification if the outcomes of this trial are indeed positive.

Reviewer: 2

Comment:

Exenatide is capitalized in some cases and in other cases not. Stay consistent throughout the

protocol.

Response:

This has now been standardized throughout the text.

Comment:

In Box 1, please clarify “levodopa equivalent doses”--- later in the protocol, it is mentioned that the LED is used to account for different dopamine replacement regimens. Perhaps stating “levodopa equivalent dose analysis” would be clearer for Box 1. Explanation of LED on page 13 is worded awkwardly. Please have a look and reword.

Response:

We have clarified the text in box 2 to say levodopa equivalent dose changes and the text in page 13 has been modified to make clearer that LED differences between groups will be adjusted for.

Comment:

Line 36 Page 9: “In the development of this protocol, a formal meeting was held with 6 patients with PD, hosted by the Cure Parkinson’s Trust.” Please clarify the role of these 6 patients. Wording needs adjusted.

Response:

Clarification on patients’ roles in the trial have now been included.

Comment:

Line 3 Page 10: Clarify what “IMP” is. It is later clarified on page 16.

Response:

This has now been defined earlier in the text as suggested.

Comment:

Line 30 Page 10: Remove parentheses

Response:

Sentence reflecting parentheses has been removed.

Comment:

BOX 2: Inclusion criteria point 1 is redundant and a copy/paste from the first paragraph of participants and recruitment. Perhaps shorten to the first sentence without mention of the Queen Square bank criteria which is mentioned in the text.

Response:

This Box includes the exact criteria which have been approved by our Research Ethics committee. We feel it is important that these are reproduced verbatim.

Comment:

Based on your previous post-hoc analysis of the data from the Phase 2 study (Athauda et al., 2019), it was found that a subset of patients were better responders to the exenatide treatment. Is any special effort being made to recruit patients based on these findings (earlier disease progression, age, etc.)? The age inclusion for the current study was increased to 80 (from 75) and the Hoehn and Yahr stage ≤ 2.5 inclusion criteria remains the same. I would be interested in any comment on this analysis and whether this analysis affected or had no effect on decisions for inclusion criteria. On page 17 there is mention of the comparative analysis between patients with lower and higher Hoehn and Yahr stage scores. Citation of (Athauda et al., 2019) would be nice here and provide some context.

Response:

We agree with the reviewer that enriching the cohort with individuals who are most likely to have the greatest response may be a useful strategy. While the effect size seen in the post hoc analysis in Athauda et al. 2019 was greatest among younger patients and those with shorter disease duration,

and indeed this is the population that tend to volunteer for participation in trials, smaller beneficial effects were also seen among more elderly patients and those with more advanced disease. We are keen to determine whether exenatide has beneficial effects on cognition, and axial features of PD, and thus took the decision to keep the inclusion criteria broad to improve our chances of detecting effects on these other outcomes and also ensuring that the results will be applicable to the broadest population of PD patients. We have added the citation as requested and a sentence to clarify this thought process.

Comment:

Line 8 Page 16:

Missing a period after citation 14.

Response:

We have ensured there is a period after citation 14.

Comment:

Line 38 Page 16: Please clarify if the 12 -week participation is the threshold for participant data inclusion. Is this a minimum participation for inclusion in analyses?

Response:

This is the threshold for inclusion into analysis. The sentence has been modified to make this clearer.

Regards,

Dr Nirosen Vijjaratnam & Prof Thomas Foltynie

VERSION 2 – REVIEW

REVIEWER	Glottelty, Elliot NIH, Drug Design and Development
REVIEW RETURNED	09-Mar-2021
GENERAL COMMENTS	Best of luck with the planned study!
REVIEWER	Banks, William Geriatrics Research Education and Clinical Center, Veterans Affairs Puget Sound Health Care System
REVIEW RETURNED	09-Mar-2021
GENERAL COMMENTS	The authors have sufficiently addressed our concerns